

# Age, adrenal steroids, and cognitive functioning in captive chimpanzees (*Pan troglodytes*)

Rafaela S.C. Takeshita[1,2,3], Melissa K. Edler[1,2,3], Richard S. Meindl[1,2], Chet C. Sherwood[4], William D. Hopkins[5] and Mary Ann Raghanti[1,2,3]

[1] Department of Anthropology, Kent State University, Kent, OH, USA
[2] School of Biomedical Sciences, Kent State University, Kent, OH, USA
[3] Brain Health Research Institute, Kent State University, Kent, OH, USA
[4] Department of Anthropology, The George Washington University, Washington, DC, USA
[5] Department of Comparative Medicine, The University of Texas MD Anderson Cancer Center, Bastrop, TX, USA

Corresponding author
Rafaela S.C. Takeshita,
rtakeshi@kent.edu

## ABSTRACT

**Background.** Dehydroepiandrosterone-sulfate is the most abundant circulating androgen in humans and other catarrhines. It is involved in several biological functions, such as testosterone production, glucocorticoid antagonist actions, neurogenesis and neuroplasticty. Although the role of dehydroepiandrosterone-sulfate (DHEAS) in cognition remains elusive, the DHEAS/cortisol ratio has been positively associated with a slower cognitive age-decline and improved mood in humans. Whether this relationship is found in nonhuman primates remains unknown.

**Methods.** We measured DHEAS and cortisol levels in serum of 107 adult chimpanzees to investigate the relationship between DHEAS levels and age. A subset of 21 chimpanzees was used to test the potential associations between DHEAS, cortisol, and DHEAS/cortisol ratio in cognitive function, taking into account age, sex, and their interactions. We tested for cognitive function using the primate cognitive test battery (PCTB) and principal component analyses to categorize cognition into three components: *spatial relationship* tasks, *tool use and social communication* tasks, and *auditory-visual sensory perception* tasks.

**Results.** DHEAS levels, but not the DHEAS/cortisol ratio, declined with age in chimpanzees. Our analyses for *spatial relationships* tasks revealed a significant, positive correlation with the DHEAS/cortisol ratio. *Tool use and social communication* had a negative relationship with age. Our data show that the DHEAS/cortisol ratio, but not DHEAS individually, is a promising predictor of spatial cognition in chimpanzees.

## INTRODUCTION

Dehydroepiandrosterone (DHEA) and its sulfated ester (DHEAS) are steroid hormones produced by the adrenal gland (*Nguyen & Conley, 2008*) as well as the gonads and the brain at smaller proportions. While they have been detected in a number of species, including birds (*Newman et al., 2008*; *Poisbleau, Lacroix & Chastel, 2009*), rodents (*Boonstra et al.,*

2008; *Quinn et al., 2013*) and marine mammals (*Gundlach et al., 2018*; *Miller et al., 2021*; *Robeck, Steinman & O'Brien, 2017*), humans and other primates are unique in having DHEA and DHEAS (hereafter denoted as DHEA(S) for both) as the most abundant circulating steroids (*Rege et al., 2019*).

Studies have demonstrated multiple biological actions of DHEA(S) (*Hildreth et al., 2013*). First, they have been associated with reproduction, as DHEA can be converted to sex steroids (*e.g.*, testosterone and estrogens) (*Labrie, 2010*; *Labrie, Martel & Balser, 2011*; *Traish et al., 2011*) and DHEAS influences oocyte maturation and predicts pregnancy outcome (*Chimote et al., 2015*; *Takeshita et al., 2016*). Second, DHEA(S) are involved in stress regulation due to their anti-glucocorticoid action (*Kalimi et al., 1994*; *McNelis et al., 2013*) by countering the neurotoxic (*Kimonides et al., 1999*) and immunosuppressive effects of glucocorticoids (*Buford & Willoughby, 2008*). During acute stress, both DHEA and DHEAS are increased (*Lennartsson et al., 2012*), but chronic exposure to stress attenuates DHEAS response (*Lennartsson et al., 2013*). Third, DHEA(S) promote neuroplasticity, neurogenesis, and neuroprotection (*Kimonides et al., 1998*), albeit they may act through different mechanisms. For instance, the neuroprotective effect of DHEA appears to occur through modulation of the calcium/nitric oxide signaling pathway, whereas DHEAS appears to act *via* the sigma-1 receptor (for reviews, see *Greaves et al., 2019*; *Maninger et al., 2009*). Similarly, the neurogenesis effect of DHEA(S) seems to be a result of the combined action of DHEA in synapse formation and axon growth with DHEAS action in enhancing dendrites (reviewed by (*Greaves et al., 2019*). In addition, several studies have reported a potential role of DHEA(S) in improving memory due to their agonist action on glutamate N-methyl-d-aspartate (NMDA) receptors (*Baulieu & Robel, 1998*; *Dong & Zheng, 2012*; *Maninger et al., 2009*; *Wen et al., 2001*).

In humans, DHEA(S) levels decline by about 20% from ages 20 to 80 years (*Vallée, Mayo & Moal, 2001*). This decline has been associated with aging processes and predisposition to diseases, including cardiovascular (*Jia et al., 2020*; *Shufelt et al., 2010*), metabolic (*Abbasi et al., 1998*; *Villareal, Holloszy & Kohrt, 2000*), and cognitive disorders (*Racchi, Balduzzi & Corsini, 2003b*; *Sorwell & Urbanski, 2010*). In humans, the aging process is associated with declines in cognitive abilities, such as processing speed, spatial memory, language, and executive function (reviewed by (*Harada, Love & Triebel, 2013*). However, there is individual variability in age-related cognitive changes, including due to medical illness, psychological factors, and sensory factors (reviewed by *Harada, Love & Triebel, 2013*). Aging is also the critical risk factor for a variety of human pathologies, including neurodegenerative diseases such as Alzheimer's disease, cancer, and metabolic diseases.

Based on the benefits of DHEA(S) in neuroprotection and its relationship with aging, DHEAS has been labeled as the "youth hormone" (*Baulieu, 1996*; *Racchi, Balduzzi & Corsini, 2003a*), and a number of clinical trials have investigated the effect of DHEA supplements to slow the aging process (*Alhaj, Massey & McAllister-Williams, 2006*; *Allolio & Arlt, 2002*; *Khorram, Vu & Yen, 1997*; *Maninger et al., 2009*; *Panjari & Davis, 2010*; *Wolkowitz et al., 1997*). However, both clinical trials and correlational studies investigating the relationship between DHEA(S) levels and cognitive function are inconclusive. While some studies showed a positive relationship between DHEA(S) and cognitive function

(*Davis et al., 2008*; *Valenti et al., 2009*; *vanNiekerk, Huppert & Herbert, 2001*), many studies show no relationship (*Barrett-Connor & Edelstein, 1994*; *Miller et al., 1998*; *Ravaglia et al., 1998*; *Yaffe et al., 1998*), and one study showed an inverse relationship (*Morrison et al., 2000*). These inconsistencies may be related to the fact that multiple intrinsic and extrinsic factors can influence these hormones and confound results. For instance, some trials with DHEA supplements were successful in improving cognition in rodents, but it was unclear if this effect was directly due to DHEA function or indirectly through its conversion to sex steroids (*Sorwell & Urbanski, 2010*). Also, several studies have found that the ratio of DHEA(S) to cortisol is a better measure of stress and psychiatric disorders. One study in soldiers reported a positive correlation between the DHEAS/cortisol and military performance under stress (*Morgan et al., 2004*), and another study revealed that the DHEAS/cortisol ratio was lower in patients under self-reported high anxiety levels (*Hartaigh et al., 2012*). Similarly, lower DHEAS to cortisol ratio has been reported in humans with dementia (*Ferrari & Magri, 2008*), depression (*Mocking et al., 2015*), and in aged rhesus monkeys (*Macaca mulatta*) exhibiting depression-like behaviors compared to age-matched controls (*Goncharova, Marenin & Oganyan, 2010*).

However, no studies have investigated the association between DHEA(S) and cognition in chimpanzees (*Pan troglodytes*). Phylogenetically, chimpanzees are one of the closest living relatives to humans, and they are known for their sophisticated cognitive skills in captivity (*Boysen & Berntson, 1995*; *Call, BA & Tomasello, 1998*; *Inoue & Matsuzawa, 2007*) and in the wild (*Boesch, Head & Robbins, 2009*; *Janmaat et al., 2014*). They also have the highest circulating DHEA(S) concentrations among nonhuman primates (*Bernstein, Sterner & Wildman, 2012*; *Rege et al., 2019*), and like humans, they experience an extended adrenarche –the postnatal secretion of these adrenal androgens (*Campbell, 2011*; *Cutler Jr et al., 1978*; *Sabbi et al., 2020*). This makes chimpanzees excellent comparative models for understanding the role of DHEA(S) in human biology and evolution.

The present study aimed to investigate the potential relationships between age, DHEAS, cortisol, as well as the DHEAS/cortisol ratio, and cognitive performance in captive chimpanzees. We predicted that the DHEAS/cortisol ratio would be a better predictor of cognitive performance than either adrenal steroid individually.

## MATERIAL AND METHODS

### Subjects and sample collection

The subjects were 107 chimpanzees (67 females and 40 males) housed in the National Chimpanzee Care Center at MD Anderson Cancer Center ($N = 77$, 48 females and 29 males) and the Yerkes National Primate Research Center ($N = 30$, 19 females and 11 males) at Emory University. Most chimpanzees were captive-born at the two facilities above. A few were wild born and imported to the U.S prior to 1974, when CITES banned the importation of chimpanzees. At the time of this project, their ages ranged from 11 to 52 years old (mean ± standard deviation (SD) = 31.5 ± 10.8 years). All chimpanzees were housed, fed and received daily enrichment according to federal regulations governing the use of nonhuman primates in research. Ten females were under oral contraception

(Provera), and nine females had intrauterine devices (IUD). One blood sample was collected per individual in the morning, during annual physical exams. During these exams, the chimpanzees were temporarily anesthetized using either ketamine or telazol, following standard operation procedures adopted at each facility. After fully recovering from the anesthesia, all chimpanzees returned to their respective social groups. This research was approved by the Institutional Animal Care and Use Committee of Emory University (Protocol nos. 2000673 and 2002189). All procedures adhered to the legal requirements of the United States and to the American Society of Primatologists' Principles for the Ethical Treatment of Primates. Blood samples were obtained prior to the Federal Register that designated the status of endangered to all captive chimpanzees under the Endangered Species Act (*U.S. Fish and Wildlife Service, 2015*).

## Cognitive tests

Subjects were tested on a modified version of the primate cognition test battery (PCTB) originally described by *Herrmann et al. (2007)* and *Herrmann et al. (2010)*. Details of the testing have been described elsewhere (*Hopkins et al., 2021*; *Russell et al., 2011*). The PCTB attempts to assess subjects' abilities in various domains of physical and social cognition. Testing was conducted between 1 to 12 years from serum sample collection (mean $\pm$ SD = 4.4 $\pm$ 2.6 years), and it was completed over one to five testing sessions, depending on the motivation and attention of the subject. All chimpanzees were given the opportunity to participate in the social and physical cognitive testing. Nine tasks were utilized in the "Physical Cognition" portion of our test battery, including tasks exploring the apes' spatial memory and understanding of spatial relationships, ability to differentiate between quantities, understanding of causality in the visual and auditory domains, and their understanding of tools. There were three tasks within the "Social Cognition" dimension of the PCTB and they are designed to assess subjects' initiation in joint attention abilities, their response to joint attention cues, and their ability to use appropriate communicative modalities based on the attentional status of a human experimenter (Attentional State).

## Hormonal assays

Serum samples were analyzed by enzyme immunoassay (EIA) developed for measurement of cortisol and DHEAS. We chose to measure DHEAS instead of DHEA, because the former is more stable and present in circulation at higher concentrations than the latter (*Kroboth et al., 1999*), and there is now evidence that peripheral DHEAS is a source of free DHEA in the brain (*Qaiser et al., 2017*). The DHEAS assay has been previously described (*Takeshita, 2022*). The cortisol assay used microplates pre-coated with a goat anti-rabbit IgG antibody (Jackson Immunoassays, Cat#111-001-003) at the concentration of 10 μg/ml, as previously described (*Khonmee et al., 2019*; *Takeshita, 2022*). The primary antibody was polyclonal anti-cortisol (BG-001) purchased from Coralie Munro (UC Davis, CA). The cortisol horseradish peroxidase (HRP) enzyme was purchased from the Endocrine Laboratory of the Smithsonian Biology Conservation Institute (Front Royal, VA). The cross-reactivities for the cortisol antibody were 100% for cortisol, 42.08% for dehydrocortisol, 26.53% for cortisone, 0.35% for corticosterone, 0.18% for desoxycorticosterone, 3.37% for prednisone, and <0.16% for tetrahydrocorticosterone.

Prior to the assay, nine standards were prepared by 1:2 serial dilutions of hydrocortisone (Cat#AAA1629203; Alfa Aesar) in assay buffer (Cat#X065; Arbor Assays) from 100 ng/g to 0.39 ng/g. The control was set at 5 ng/g. Following standard preparation, serum samples were diluted at 1:10 (cortisol) in assay buffer and taken to the EIA following the procedures previously described (*Takeshita, 2022*) with minor adaptations. In brief, 50 μl of samples, standards and controls were added to each designated well in duplicate. Assay buffer was added to non-specific binding (NSB) (75 μl) and B0 wells (50 μl), also in duplicate. In sequence, 25 μl of cortisol HRP diluted in assay buffer (1:5,000) were added to each well. Immediately after adding HRP, 25 μl of anti-cortisol diluted in assay buffer (1:25,000) were added to each well, except NSB wells. The plates were sealed and incubated at room temperature for 1 h (cortisol assay). After the incubation time, the microplates were washed four times with wash buffer (0.5% Tween-20, 1.5 M sodium chloride), blotted dry and developed by adding 100 μl of 60% High-kinetic TMB (TMBHK60; Moss Inc., Franklin Park, IL, USA) to each well, followed by incubation in the dark at room temperature for 10 min. The reaction was stopped by adding 50 μl of stop solution (1N HCl) to each well, and the plate was read in a plate reader (BioTek TSI 800; BioTek, Winooski, VT, USA) at 450 nm.

To validate the two hormonal assays for chimpanzee serum, parallelism tests were conducted by serially diluting a pooled sample in assay buffer from 1:2 to 1:64 for the cortisol and from 1:2 to 1:512 for the DHEAS assay, due to the high concentration of this steroid in the samples. The curves generated by the serially diluted pooled samples were visually inspected for parallelism with the standard curves in each hormonal assay and confirmed by F-tests. Both visual inspection and F-tests indicated parallelism for cortisol ($F_{8,8} = 0.51$, $p = 0.40$) and DHEAS assays ($F_{10,8} = 0.99$, $p = 0.90$). Additionally, accuracy tests were conducted by spiking a pooled sample with known amounts of steroids and measured using the EIAs described above. The mean ± SD recoveries were 85.9 ± 3.4% for cortisol and 98.1 ± 6.3% for DHEAS. The successful parallelism and accuracy tests indicated that the assays were considered suitable for chimpanzee serum, so we analyzed all samples in duplicate. The intra-assay coefficients of variation (CV) ($N = 107$) were 3.6% and 5.16%, and the inter-assay CVs ($N = 4$) were 11.6% and 12% for cortisol and DHEAS assays, respectively.

## Statistical analyses

We used R software version 4.1.0 (*R Core Team, 2017*) for the regression analyses and IBM SPSS Statistics for Windows version x.0 (SPSS Inc. Chicago, USA), licensed for Kent State University, for the principal component analyses (PCA). To exclude the possibility of hormonal contraception as a confounding factor in the hormonal analyses, we first built two linear models with only females ($N = 67$) to test the effect of hormonal contraception (fixed factor) on DHEAS and the DHEAS/cortisol ratio (response factors), controlling for age. Normality was confirmed visually by diagnostic plots (histogram of frequency, quantile–quantile plot, distribution of residuals) and Shapiro–Wilk normality test (*Shapiro & Wilk, 1965*). Homoscedasticity across categorical factors (contraception) was confirmed by Levene's test (*Levene, 1960*). First, an initial model including all fixed effects was

tested for multicollinearity by calculating variance inflation factors (VIF) of the fixed factors with the package *car*. All factors with VIF >2 were considered problematic and were removed from the model. If the model distribution was not normal, we applied power transformation using the *boxcox* function from the package *MASS* (*Box & Cox, 1964*). Following *Burnham & Anderson (2002)*, we initially build a full model including all predictors and their interactions, then sequentially removed fixed factors with the highest *p*-value. The same principle was applied to polynomial models (cubic and quadratic). We selected the models with the lowest Akaike Information Criterion (AIC). Contracepted females were not significantly different than non-contracepted females in any of the models (DHEAS $\beta \pm$ SE: $0.51 \pm 0.79$, $t = 0.65$, $p = 0.52$; DHEAS/cortisol ratio $\beta \pm$ SE: $-0.01 \pm 0.04$, $t = -0.1$, $p = 0.92$), so we included all individuals in our subsequent analyses.

To account for potential effects of age, sex or colony differences in hormonal levels, we first tested cortisol, DHEAS and the DHEAS/cortisol ratio as response factors, with individual age during serum sample collection, sex, and colony (Yerkes or MDACC), and their interactions using linear and polynomial models as described above. In the cortisol model, colony showed heteroscedasticity (Levene's test: $F_{1,105} = 6.00$, $p = 0.01$), which was corrected after the removal of one outlier (Levene's test: $F_{1,104} = 3.78$, $p = 0.06$). In both DHEAS and cortisol models, the response factor was power-transformed to meet the assumptions of normality of residuals.

To reduce the dimensionality of the cognitive data, we conducted PCA on the intercorrelations of 12 individual cognitive tasks assessed during the PCTB. The unrotated solution yielded five components with eigenvalues >1, which explained between 14.54% and 9.59% of the variance. However, components 4 and 5 did not yield much new information on any variables, with only one or two having high loadings (>0.5). We further reduced the analysis to three components, and using Varimax rotation with Kaiser normalization, all 12 variables were adequately represented. The three components that were extracted explained 14.76%, 13.63%, and 12.97% of the variance, respectively. PC1 (*spatial relationships*) included the tasks "spatial memory", "object permanence", "rotation", and "transposition". PC2 (*tool use and social communication*) included the tasks "tool use success", "tool properties", "gaze following", "initiation of joint attention (gesture production)", and "attention state". PC3 (*auditory and visual sensory perception*) included "relative numbers", "causality noise", and "causality visual" tasks. Using the Z-values from these three components, we built linear and polynomial models to test the effect of DHEAS, DHEAS/cortisol ratio, sex and age in each PC as described above. Due to a significant linear effect of age in DHEAS levels (see results), we excluded 87 individuals which had >1 year difference between serum sample collection and PCTB testing in the subsequent analyses (but see Supplementary Material for additional analyses with the full dataset for comparative purposes). Considering the small sample size in these models ($N = 21$, 15 females and 6 males), we used the AIC with correction for small sample size (AICc) to select the best model (function *model.sel* from the package *MuMIn*). All models not different than the best model ($\Delta$AICc <2) were discussed. All plots were generated using the package ggplot2 (*Wickham, 2009*).

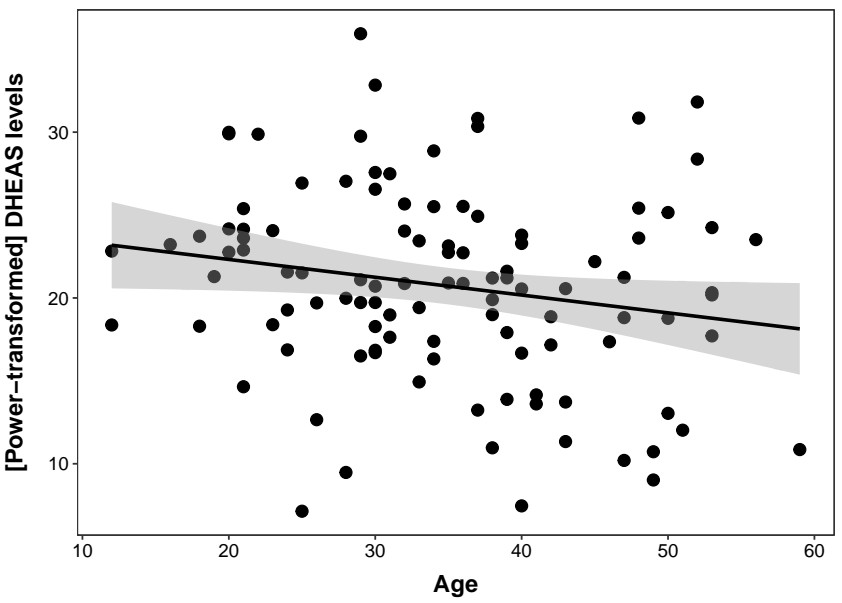

**Figure 1** **Relationship between serum DHEAS levels and age in 107 captive chimpanzees (*Pan troglodytes*).** Each data point represents one individual. Data on DHEAS levels were power-transformed to fit model assumptions. The regression line represents the predicted relationship between DHEAS levels and age, and the shaded area represents a 95% confidence interval on the fitted values.

## RESULTS

The DHEAS/cortisol ratio models were not better than the null model, indicating no relationship between this hormonal index and age, sex, or colony. The best cortisol model had a non-significant effect of age ($\beta \pm \text{SE} = -0.03 \pm 0.29$, t $= -1.82$, $p = 0.07$), even when controlling for sex and colony, and this model was not different than the null model ($\Delta\text{AIC} = 1.3$). The DHEAS model revealed a negative relationship between DHEAS levels and age ($\beta \pm \text{SE} = -0.1 \pm 0.05$, t $= -2.08$, $p = 0.04$), which explained 4% of the variation in the data (Fig. 1), controlling for sex and colony. Polynomial models were not a better fit for either cortisol, DHEAS or the DHEAS/cortisol ratio data. Colony was not significant in any of the models, so to reduce model complexity, this factor was excluded from the following models.

In the PCTB models including age, sex, cortisol, DHEAS and the DHEAS/cortisol ratio, there was a multicollinearity between cortisol and the DHEAS/cortisol ratio (VIF $= 3.15$ and 2.24, respectively). For this reason, cortisol was removed from the model and analyzed separately with age and sex, to test the possible effect of cortisol individually on cognition. The best PCTB PC1 (*spatial relationships*) model included a significant, linear, and positive relationship between PC1 and the DHEAS/cortisol ratio ($\beta \pm \text{SE} = 0.03 \pm 0.01$, $t = 2.35$, $p = 0.03$), which explained 22% of the variance in PC1 (Fig. 2). Sex, age, and DHEAS levels alone were not significant and did not improve the model. The second model that included cortisol as the only steroid hormone was not better than the null model, indicating that this steroid alone did not affect PC1.
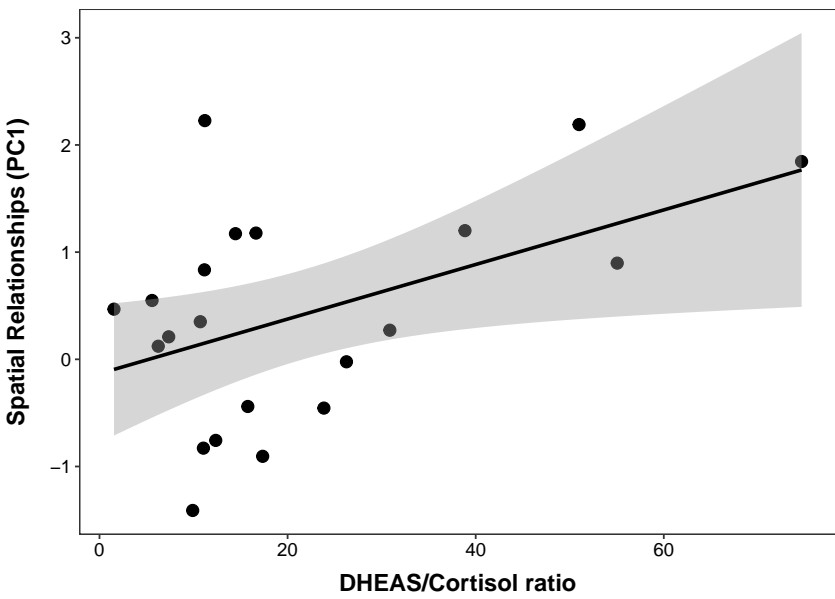

**Figure 2 Positive correlation between DHEAS/cortisol ratio and Primate Cognition Testing Battery PC1 (spatial relationships) in chimpanzees.** Each data point represents one individual ($N = 21$). The regression lines represent the predicted relationship between PC1 and the DHEAS/cortisol ratio. The shaded areas represent a 95% confidence interval on the fitted values.

The best PCTB PC2 (*tool use and social communication*) model included a significant linear effect of age ($\beta \pm$ SE $= -0.06 \pm 0.02$, t $= -3.40$, $p = 0.003$), but no effect of DHEAS/cortisol ratio, sex, DHEAS, or their interactions. In this model, age explained 38% of the variance in PC 2 (Fig. 3). The second model that included cortisol as the only steroid hormone resulted in the same reduced model described above, with age as the only fixed factor. This indicates that none of the adrenal hormones influenced this component.

The best PCTB PC3 (*auditory and visual sensory perception*) model included a non-significant effect of sex ($\beta \pm$ SE $= 0.45 \pm 0.46, t = 0.98, p = 0.34$) and it was not significantly different than the null model ($\Delta$AICc $= 1.7$). Similarly, the second model that included cortisol as the only steroid hormone was not better than the null model. Thus, none of the predictors influenced PC3. Polynomial models were not a better fit for any of the response variables.

## DISCUSSION

The present study tested the relationship between age, DHEAS, cortisol, and the DHEAS/cortisol ratio with chimpanzee cognitive performance. We found that contraception, sex, and colony did not affect hormonal levels, but age was negatively correlated with DHEAS levels. We reduced our cognitive data derived from performance on the PCTB to three principal components. The first component (PC1) reflected individual chimpanzees' performance on spatial relationships tasks, and we found a significant, positive relationship between DHEAS/cortisol ratio and this component. The
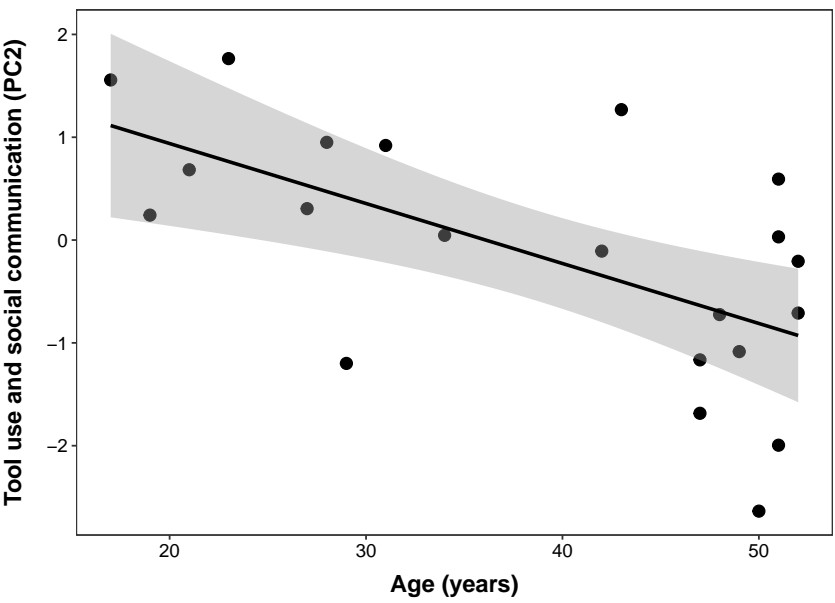

**Figure 3** Negative correlation between age and Primate Cognition Testing Battery PC2 (tool use and social communication) in chimpanzees. Each data point represents one individual ($N = 21$). The regression lines represent the predicted relationship between PC2 and age. The shaded areas represent a 95% confidence interval on the fitted values.

second component (PC2) reflected chimpanzees' tool use and social communication abilities. We found a significant negative effect of age in this component. Finally, the third component (PC3) quantified their abilities to discriminate quantity and understand causal relationships. We found no significant effects of age, sex, hormones, or their interactions on this component.

The negative correlation between age and DHEAS levels observed in the present study is consistent with previous studies in rhesus macaques (*Muehlenbein et al., 2003*), Japanese macaques (*Takeshita et al., 2013*) and lemurs (*Perret & Aujard, 2005*) that show an age-related decline in DHEAS levels. One cross-sectional study in chimpanzees reported a modest age-related decline in female chimpanzees from 15 to 54 years old (*Blevins et al., 2013*), and our findings extend this pattern to male chimpanzees. A recent longitudinal study using urine samples from wild chimpanzees showed that DHEAS levels start to rise from 2-3 years of age until adulthood, which is a period known as adrenarche, with no sex differences in hormonal levels (*Sabbi et al., 2020*). We did not observe an increase in DHEAS levels, even during early adulthood, possibly due to our small sample size of individuals below 20 years of age ($N = 10$). However, our results showed no sex differences in serum DHEAS levels, which was consistent with *Sabbi et al. (2020)*. In humans, aging has been associated with the thinning of the zona reticularis and a relative increase in zona glomerulosa and fasciculata, which may explain the age-related decline in DHEAS secretion (*Parker Jr et al., 1997*).

We also found that contraception did not influence DHEAS levels, which supports previous studies in chimpanzees (*Blevins et al., 2013*), ovariectomized rhesus macaques (*Conley et al., 2013*), and long-tailed macaques (*Henderson & Shively, 2004*). In contrast, studies in humans showed that contraception decreased DHEAS levels in women (*Enea et al., 2009*), but that these changes are related to alterations in serum albumin, DHEAS' main binding protein (*Carlström, Karlsson & Schoultz, 2002*; *Panzer et al., 2006*). In premenopausal women, approximately 50–75% of circulating estrogen derive from adrenal androgens, while in post-menopausal women, this rate is estimated to 100% (*Labrie et al., 1998*; *Samaras et al., 2013*). However, there is still a debate on whether chimpanzees experience menopause and at what age (*Ellis et al., 2018*; *Thompson et al., 2007*), so it is possible that the influence of DHEAS in female reproduction may differ between these species. In addition, the effect of oral contraception in DHEAS levels can be affected by age (*Conley et al., 2013*) and by the type of contraceptive (*Trienekens, Schmidt & Thijssen, 1986*), which may also explain the contrast between these studies.

Regarding the cognitive tests, we found a positive relationship between the DHEAS/cortisol ratio and PC1 factor, which indicates that a high DHEAS to cortisol ratio is associated with better cognitive performance. The comparative analyses with the full dataset (Supplementary Material) revealed an interaction between age and the DHEAS/cortisol ratio in this component, which suggested that the positive correlation between the DHEAS/cortisol ratio and PC1 was more evident or exclusive to elder chimpanzees. However, this interaction could be due to noise in the data resulting from the discrepancy between age at serum sample collection and at PCTB testing, which spanned up to 12 years. For this reason, the discussion focused on the results from the reduced dataset (controlled for age discrepancy).

Previous studies in humans have associated low DHEAS levels in elderly humans with age-related conditions, including cognitive decline (*Bologa, Sharma & Roberts, 1987*; *Flood & Roberts, 1988*; *Flood, Smith & Roberts, 1988*; *Moffat et al., 2000*), cardiovascular diseases (*Jia et al., 2020*; *Shufelt et al., 2010*), and Alzheimer's disease (*Weill-Engerer et al., 2002*). DHEAS has its affinity to sigma-1 receptors, and acts by facilitating neurotransmission in hippocampal neurons and NMDA signaling (*Yabuki et al., 2015*; *Yoon et al., 2010*), which play a critical role in spatial memory (*Bettio, Rajendran & Gil-Mohapel, 2017*). Experimental studies have shown that DHEAS improves memory retention in rodents (*Flood & Roberts, 1988*) and acts in the neocortex and hippocampus by increasing NMDA receptors, a glutamate receptor involved in neural plasticity and cognitive processes (*Collingridge et al., 2013*; *Wen et al., 2001*). Our findings suggest that the ratio of DHEAS to cortisol is one possible mechanism mediating cognitive changes in chimpanzees. In humans, a high cortisol to DHEAS ratio has been negatively correlated with hippocampal, amygdala, and insula volume in humans, and with tau and p-tau levels (*Jin et al., 2016*; *Ouanes et al., 2022*). Although further research correlating brain volume and adrenal steroids in chimpanzees is needed to test whether this relationship is similar in chimpanzees and humans, a previous study that included the chimpanzees described in the present work found age-related loss in grey matter in brain regions associated with cognition, and old chimpanzees with greater grey matter performed better within their age category

(*Mulholland et al., 2021*). However, we did not find an aging effect of spatial cognition tests, which could be due to a statistical power issue, but it might indicate that the DHEAS/cortisol ratio was a better explanatory factor for variation in this cognitive component.

The fact that PC1 was associated with the DHEAS/cortisol ratio, but not DHEAS or cortisol levels independently, may explain why the literature on the relationship between adrenal steroids and cognition is inconsistent (*Sorwell & Urbanski, 2010*; *Vallée, Mayo & Moal, 2001*). Previous studies have reported that DHEAS levels are affected by acute and chronic stress due to its antagonistic action on glucocorticoids (*Kalimi et al., 1994*; *Maninger et al., 2010*; *McNelis et al., 2013*). For this reason, recent studies that adopted the co-measurement of cortisol and DHEAS to investigate cognition and stress levels in several species, including marine mammals (*Gundlach et al., 2018*; *O'Brien et al., 2017*), ungulates (*Almeida et al., 2008*; *Jurkovich et al., 2020*), humans (*DeBruin et al., 2002*; *Miller et al., 1998*; *Ouanes et al., 2022*), and nonhuman primates (*Goncharova, Vengerin & Chigarova, 2012*; *Maninger et al., 2010*; *Takeshita et al., 2014*; *Takeshita et al., 2019*).

Our findings may also clarify why there is usually an inverted U-relationship between stress and cognitive performance (*Sapolsky, 2015*). One study in rhesus monkeys demonstrated that moderate stress stimulates DHEAS production (*Goncharova, Vengerin & Chigarova, 2012*). By competing with cortisol for glucocorticoid receptors (GR), higher DHEAS availability will prevent the deleterious effects of cortisol in the brain that are associated with the binding of cortisol to GR. However, intense or prolonged stress will result in a decrease in the DHEAS/cortisol ratio due to the continuous stress stimuli (*Sugaya et al., 2015*). Higher cortisol to DHEAS binding of GR could promote neurodegeneration, which negatively affects memory and cognition (*deKloet, Oitzl & Joels, 1999*). Due to the competitive relationship between DHEAS and cortisol, the use of both hormones in stress studies is a better indicator of the DHEAS availability than is either hormone measured alone (*Gabai et al., 2020*; *Whitham, Bryant & Miller, 2020*).

Hormonal levels were not correlated with PC2, but we found a negative correlation between PC2 and age. In contrast, PC3 was not associated with age nor with hormonal levels with the reduced sample size. Although further analyses with the full data set indicated a negative effect of age and cortisol on this component, the effect of age disappeared when cortisol was removed from the model, and vice-versa (Supplementary Material). The inconsistency in the models and small effect size of age (2.9%) suggest that this result is likely not biologically significant.

Age-related decline in cognition has been widely reported in humans and other primates (*Hara, PR & Morrison, 2012*; *Herndon et al., 1997*; *Hopkins et al., 2021*; *Lacreuse et al., 2018*; *Lacreuse et al., 2014*; *Rothwell et al., 2022*), and it has been associated with cortical thinning (*Ahn et al., 2011*), grey matter atrophy (*Mulholland et al., 2021*; *Nickl-Jockschat et al., 2012*), a decline in neuron density (*Edler et al., 2020*; *Hara, PR & Morrison, 2012*; *Wilson et al., 2010*), oxidative stress, neuroinflammation, and altered hippocampal intracellular signaling and gene expression (reviewed by (*Bettio, Rajendran & Gil-Mohapel, 2017*). Longitudinal studies investigating cognitive decline in chimpanzees have reported that the aging effect is more pronounced in older individuals performing spatial tasks, which agrees with our findings (*Hopkins et al., 2021*). The lack of a hormonal or aging effect on PC3 in

comparison to PC1 and PC2, respectively, suggests that chimpanzees are more susceptible to a performance decline in tasks requiring spatial memory or social communication skills in comparison to audio-visual sensory perception.

One limitation of this study is that our data are cross-sectional, and there are inter-individual differences that may affect test performance or hormonal levels. However, considering animal research ethics and the classification of captive chimpanzees as endangered, longitudinal data on chimpanzee serum paired with cognitive data are difficult to obtain in sufficiently large numbers. Nevertheless, our findings revealed important characterizations between adrenal steroids and aging in chimpanzees. First, we demonstrated that DHEAS declines with aging in both males and females. Second, we showed evidence that the DHEAS/cortisol ratio correlated with spatial cognition in chimpanzees. Chimpanzees have an extended postnatal increase in DHEAS levels called adrenarche, which appears to be unique to humans and great apes and spans from the pre-pubertal period to mid-adulthood (*Bernstein, Sterner & Wildman, 2012*; *Copeland et al., 1985*; *Cutler Jr et al., 1978*; *Sabbi et al., 2020*). Although the reasons for the emergence of adrenarche in hominids is still unclear, current hypotheses suggest that this trait evolved to promote brain development during early growth in both humans and great apes (*Campbell, 2020*; *Campbell, 2021*). Our findings are consistent with this hypothesis and further suggest that the extended adrenarche in these species might have contributed to a prolonged period of heightened DHEAS levels, which may buffer the age-related cognitive decline in these species. Based on evidence on the function of DHEAS in neuroprotection and neuroplasticity (*Bastianetto et al., 1999*; *Dong & Zheng, 2012*; *Flood & Roberts, 1988*; *Kimonides et al., 1998*; *Kimonides et al., 1999*; *Majewska, 1995*) and the similarities between humans and chimpanzee with regards to adrenal androgen secretion patterns (*Bernstein, Sterner & Wildman, 2012*; *Rege et al., 2019*), we hypothesize that DHEAS contributed to human cognitive evolution.

In summary, our study is the first to investigate the relationship between DHEAS/cortisol ratio and cognition in chimpanzees. Our data show evidence of a positive correlation between DHEAS/cortisol ratio and spatial cognition in chimpanzees and of an aging effect on tool use and social communication. These results contribute to our understanding of the role of DHEAS in human evolution and highlight the importance of integrating cortisol and DHEAS in the investigation of age-related disorders.

## ACKNOWLEDGEMENTS

We thank Mary Ann Cree and Brenda Webb for their assistance with serum sample collection.

### Funding

This study was funded by the National Institute of Health (No. R01AG067419), by the National Science Foundation (No. 2123574), and by the Wenner-Gren Foundation (No.

10088). The funders had no role in study design, data collection and analysis, decision to publish, or preparation of the manuscript.

## Grant Disclosures

The following grant information was disclosed by the authors:

The National Institute of Health: R01AG067419.

The National Science Foundation: 2123574.

The Wenner-Gren Foundation: 10088.

## Competing Interests

The authors declare there are no competing interests.

## Author Contributions

- Rafaela S.C. Takeshita conceived and designed the experiments, performed the experiments, analyzed the data, prepared figures and/or tables, authored or reviewed drafts of the article, and approved the final draft.
- Melissa K. Edler analyzed the data, authored or reviewed drafts of the article, and approved the final draft.
- Richard S. Meindl analyzed the data, authored or reviewed drafts of the article, and approved the final draft.
- Chet C. Sherwood conceived and designed the experiments, authored or reviewed drafts of the article, and approved the final draft.
- William D. Hopkins conceived and designed the experiments, performed the experiments, authored or reviewed drafts of the article, and approved the final draft.
- Mary Ann Raghanti conceived and designed the experiments, performed the experiments, analyzed the data, authored or reviewed drafts of the article, and approved the final draft.

## Animal Ethics

The following information was supplied relating to ethical approvals (i.e., approving body and any reference numbers):

The Institutional Animal Care and Use Committee of Emory University approved the study (Protocol #2000673 entitled "Neural Imaging pf Primates"; Protocol #2002189 entitled "Hemispheric specialization an and Communicative and Social Competency in Chimpanzees").

## Data Availability

The raw data and R code used for the statistical analyses is available in the Supplemental Files.

## Supplemental Information

Supplemental information for this article can be found online at http://dx.doi.org/10.7717/peerj.14323#supplemental-information.

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
