# Peer review of "Age, adrenal steroids, and cognitive functioning in captive chimpanzees (Pan troglodytes)"

_PeerJ, doi:10.7717/peerj.14323_

## Round 0.1 · original submission · Major Revisions

Your article entitled “Age, adrenal steroids and cognitive functioning in captive chimpanzees (Pan troglodytes)” has now been seen by two reviewers and the reviewers’ comments are appended below. I share the reviewers' view that this manuscript provides an interesting first step in describing the potential hormonal correlates of cognitive functioning. However, the concerns they are also raising mean that I do not think that this article is at this stage ready for publication. For me, there are two points that both reviewers mention as "crucial", which they appear to me to be, which is why this article needs more than a minor revision. All the comments are about providing more explanations of the underlying rationale of your study, on what was done and why, and more clearly (for example already in the abstract) mentioning the limitations of what can be inferred. I think these data have the potential to provide interesting insights, but, based on the information provided in the current manuscript, I am not convinced that you can make the inferences you provide.

First point: time between data collection of hormonal and cognitive data
As reviewer 1 says, there “is a major potential issue which you attempt to control for, but more detail on how and why this factor (which ultimately is eliminated from later models) is sufficient to control for this concern is needed."
You cite some studies on age-related changes in the levels of these hormones. Do these studies actually report that inter-individual differences are stable over time? This seems what you are assuming here. Another reason for why the DHEAS/cortisol ratio was not influenced by age in your dataset could be that these hormones (cortisol in particular) vary depending on time of day/stress/etc. which differed among the individuals from whom you collected the data. You argue that you controlled for stress by controlling for cortisol. Wouldn’t that mean that the DHEAS/cortisol ratio is therefore not a reflection of stress? However, in your discussion, you mainly talk about the link between DHEAS, stress, and cognition. What do you think the single hormonal levels you measured years ago actually reflect? As an outsider, I thought that both these hormones can change even in a matter of hours, so I am not sure how a single measure years ago would link to cognitive abilities. That might simply be my lack of understanding, but I think it would help other readers if you explained this more clearly.

Second point: how do you think the hormones link to the cognitive abilities
Linked to the point above, your introduction does not make it clear how you expect your particular hormonal measurements to link to cognitive abilities years later. Most of your discussion is a post-hoc interpretation of the patterns you found. This makes the study rather descriptive, which is fine, but it needs extra caution to understand the limitations and the avoidance of any overinterpretation (such as in your abstract where you claim that “Our data show that the DHEAS/cortisol ratio, but not DHEAS individually, is a promising predictor of age-related cognitive decline in chimpanzees and may be involved in spatial cognition.”).
Reviewer 2 refers to this when asking about the discussion between potential connections with gray matter/cortical thickness and other brain changes.
I think in general it would help if you specify more clearly how you think the different factors in your model influence each other. For example, from a point of understanding, I am incapable of reconstructing what a three way interaction means without a clear specification of what these interactions would be. On a broader note, without a careful model, introducing additional factors introduces the risk of introducing a confound that might obscure the actual relationships. Here is an article that discusses some of these issues and might serve as a starting point: “Yes, But What’s the Mechanism? (Don’t Expect an Easy Answer)” DOI: 10.1037/a0018933 https://www2.psych.ubc.ca/~schaller/528Readings/BullockGreenHa2010.pdf
For an explanation of causal models, and in particular why AIC might not help, you might want so see this course: https://github.com/rmcelreath/causal_salad_2021
For example, you apparently include “DHEAS, corrected by age” plus “age” itself in the same model. I am not sure what relationship with age you expect when you already corrected for age. Also, if you corrected one predictor variable for age, you might introduce confounds (see material above).
More specifically, for the main result you focus on, you report that “DHEAS/cortisol ratio is more important for cognition in elderly chimpanzees”. However, as far as I can see, that is not what your figure 2 shows. Your figure 2 seems to indicate that the DHEAS/cortisol ratio appears to correlate with spatial cognition in all ages, simply in opposite directions. Again, here it would be helpful to have a mechanistic explanation: why would you expect that young chimpanzees with a high DHEAS/cortisol ratio are worse at spatial cognition than young chimpanzees with a low DHEAS/cortisol ratio? And how do you interpret this interaction to mean that “the DHEAS/cortisol ratio is a promising predictor of age-related cognitive decline in chimpanzees” when you did not find an age related decline in spatial cognition. In your model, age did not appear to have a direct effect on spatial cognition and figure 2 shows that the median/mean values for all three age groups is the same (and the individual with the best performance appears to be one of the oldest chimpanzees). This would indicate that there is no age related decline in this cognitive ability, so your data and analyses cannot make any inferences about how hormones might link to age related declines. While the other two cognitive abilities appear to show a decline with age, apparently here the hormones are not involved, so again this would argue against your statement that these hormones are promising predictors of age-related cognitive decline. Again, without an underlying mechanistic model, it is impossible to interpret what an interaction in a statistical model means. In a case such as here where we might not yet have sufficient studies to build mechanistic models, plotting the data and reporting that findings from there might be the better approach.

Reviewer 1 ·

Basic reporting

Takeshita et al. present analyses of proposed hormonal correlates of aging and their effect on cognitive performance. The study is novel, interesting, and well-powered to address these questions. The background and discussion were largely relevant and appropriate, and the hormonal analyses thoroughly described. However, more details are needed in some areas. In the introduction, there is some jumping between DHEA(S) and DHEAS. The authors report that they choose to focus on DHEAS in their analyses due to its greater stability, but the introduction should be more careful about distinguishing which compound they refer to rather than treating them initially as interchangeable. More information on their relation and which were the focus of the empirical work reviewed in the introduction would improve the clarity of the paper. In the results, some statistics require further reporting. In the paragraph beginning on line 239, models results are reported which do not connect with the model descriptions in the methods (which are focused on only the effect of contraception and the effect on PCA results). The inclusion of the 3-way interaction between age at cognitive testing, hormone levels, and difference between ages at the two parts of the study requires more detailed justification. This is a major potential issue which you attempt to control for, but more detail on how and why this factor (which ultimately is eliminated from later models) is sufficient to control for this concern is needed.

Experimental design

The overall experimental design was straightforward and intuitive. I have just a few small questions and suggestions. Did the authors consider including cortisol itself as a predictor in their models? While they find DHEAS/cortisol ratio, but not DHEAS itself, interact with age in predicting cognitive performance, an intuitive hypothesis is that cortisol itself may be causing some of the effect without needing to interact with DHEAS. The role of cortisol on its own without relating to DHEAS should be investigated. It might also be worthwhile to confirm whether there are basic effects of age/sex/colony on cortisol in your data as you did with DHEAS.
Given your explanation for the interaction effect between age and DHEAS/cortisol ratio in predicting PC1 performance, and the graph dividing the age categories into 3, I wonder if it would support your results to run analyses on the 3 age categories separately in follow-up models. This would strengthen your claim that the interaction is due to an effect in elderly, but not younger, chimpanzees.

Validity of the findings

The results were largely clean and well described, and placed into the existing literature.
My only major concern relates to the point made earlier about controlling for age differences between serum collection and cognitive testing. There is some mention of this in the discussion, and inclusion of an interaction effect in models to attempt to remedy this possible issue, but much more detail and thorough reasoning is required. In my opinion this is the major factor threatening the interpretability of the results, so I recommend adding significantly more discussion of the authors' reasoning and approach to this. More focused statistical models to address this question beyond the inclusion of a 3-way interaction effect may be warranted. Some numeric information about the average time between these two events would also be helpful. If this issue can be convincingly resolved I see no other issues of validity.

Additional comments

Some small line by line comments:
Line 26: In abstract add short form DHEAS in parentheses after the full name
Line 67 indicate that this result is from humans
Line 73: add the words “due to” in the phrase "including *due to* mental illness, ..."
Line 76-77: this sentence is a bit confusing as written
Line 94: you here use cortisol/DHEAS, but use DHEAS/cortisol everywhere else. Please check which is correct, and consider rephrasing so you can use the same ratio each time
Lines 204-217: Were the PCA factors found here consistent with previous PCA factors on data from the great ape cognition test battery?
Line 259: indicate which direction of effect in the paragraph
Line 263: it is already implied, but might be helpful to explicitly state DHEAS was not involved at all
Line 300: chimpanzee do not typically experience menopause, but old chimpanzees can, so this phrasing should be a bit careful (see for example Herndon et al. 2012, DOI: 10.1007/s11357-011-9351-0)
Line 382: there's a weird citation formatting
Line 384: your results are consistent with, but don’t necessarily support this hypothesis I don't think

·

Basic reporting

Please see attachment.

Experimental design

Please see attachment

Validity of the findings

Please see attachment

Additional comments

Please see attachment.

---

## Round 0.2 · Minor Revisions

Thank you for addressing the comments so diligently and for making the respective changes in the manuscript. In particular, the changes to the analyses now provide clearer insights into the questions you study. However, I also agree with the reviewer that there are some parts now where the reporting does not seem to fully match these results. The reviewer highlights the final sentence of the abstract, where you make a statement even though your results do not support a link between DHEAS and cognitive changes with age. There are similar issues in the discussion, where you for example state that “Our results suggest that, in addition to age-related decline in grey matter volume, the effect of aging in spatial cognition may be mediated by adrenal steroids” (there are more places where you focus on this link). Based on my reading, none of your results support this. The DHEAS levels, the only hormonal measure for which you find an age-related change, is not associated with the performance in any of the cognitive tasks, whereas the DHEAS/cortisol ratio does not change with age. Even the interaction you now moved to the supplementary materials does not show the predicted relationship between a change in hormones over age that matches a change in cognitive performance. The negative result could be due to a power issue, but alternatively, it could be that your results actually contradict your statements. I would therefore ask you to change or remove these parts to accurately reflect your results. In terms of matching the results to the remaining manuscript, I also noticed that in the final part of the introduction (Lines 145-148) where you introduce the aims of your study, age is not mentioned. You might want to expand this section to introduce all the analyses. I think these findings are interesting in their own right and will be useful for future studies, and I look forward to seeing the revised manuscript.

Reviewer 1 ·

Basic reporting

I am pleased to see the authors carefully considered and thoroughly replied to all the comments from myself and the other reviewer, and revised the manuscript appropriately.

Experimental design

no comment

Validity of the findings

no comment

Additional comments

My only remaining concern is the final line in the abstract which reads "Our data show
that the DHEAS/cortisol ratio, but not DHEAS individually, is a promising predictor of age-
related cognitive decline in chimpanzees and may be involved in spatial cognition." The results find DHEAS itself declined with age, while the ratio was associated with spatial cognition. Neither was associated with both aging and cognition (but both do seem promising for future research on the topic), so this final sentence should be slightly modified to reflect that.

---

## Round 0.3 · accepted · Accept

Thank you for making these final changes. I read through the revised version, which is now ready for publication.